# AstroAgents: A Multi-Agent AI for Hypothesis Generation from Mass Spectrometry Data

## Abstract

With upcoming sample return missions across the solar system and the increasing availability of mass spectrometry data, there is an urgent need for methods that analyze such data within the context of existing astrobiology literature and generate plausible hypotheses regarding the emergence of life on Earth. Hypothesis generation from mass spectrometry data is challenging due to factors such as environmental contaminants, the complexity of spectral peaks, and difficulties in cross-matching these peaks with prior studies. To address these challenges, we introduce *AstroAgents*, a large language model-based, multi-agent AI system for hypothesis generation from mass spectrometry data. *AstroAgents* is structured around eight collaborative agents: a data analyst, a planner, three domain scientists, an accumulator, a literature reviewer, and a critic. The system processes mass spectrometry data alongside user-provided research papers. The data analyst interprets the data, and the planner delegates specific segments to the scientist agents for in-depth exploration. The accumulator then collects and deduplicates the generated hypotheses, and the literature reviewer identifies relevant literature using Semantic Scholar. Finally, the critic evaluates the hypotheses, offering rigorous suggestions for improvement. To assess *AstroAgents*, an astrobiology expert evaluated the novelty and plausibility of more than a hundred hypotheses generated from data obtained from eight meteorites and ten soil samples. Of these hypotheses, surprisingly, 36% were identified as plausible, and among those, 66% were novel.

## 1 Introduction

The rapid growth of spectrometry data from sample return missions the solar system where traces of past, extinct, or present life can be found necessitates methods to analyze this massive, high-dimensional data and generate plausible hypothesis on one of the most fundamental questions in astrobiology: How did life emerge on Earth? Lahav et al. (2001); Pross & Pascal (2013) Analyzing mass spectrometry data in astrobiology is challenged by the presence of terrestrial contaminants Glavin et al. (2025), the complexity of spectral peaks, and the lack of a systematic approach for hypothesis generation by comparing and contrasting to existing mass spectrometry data Kitano (2021). Hypothesis generation by human experts is often biased, time-consuming, and limited to the literature that the individual has expertise in Nissen et al. (2016). Computational methods, on the other hand, are challenged by the sparsity of peaks relevant to the dimension of the mass spectrometry data, which makes identifying patterns extremely difficult Guo et al. (2022).

Recent advances in large language models (LLMs) have demonstrated remarkable capabilities in scientific reasoning Truhn et al. (2023) and hypothesis generation Zhou et al. (2024); Zhang et al. (2024). However, these models face inherent limitations when deployed individually: they struggle with consistent reasoning over complex datasets, lack specialized domain expertise, and cannot independently validate their outputs against scientific literature Kaddour et al. (2023). These limitations become particularly apparent in origins of life research, where hypotheses must bridge multiple disciplines and incorporate complex molecular evidence from mass spectrometry data.

Multi-agent architectures have emerged as a promising approach to overcome the limitations of LLMs. Recent work has shown how multiple AI agents, each with specialized roles, can collaborate to enhance scientific discovery. *SciAgents* Ghafarollahi & Buehler (2024), a multi-agent AI

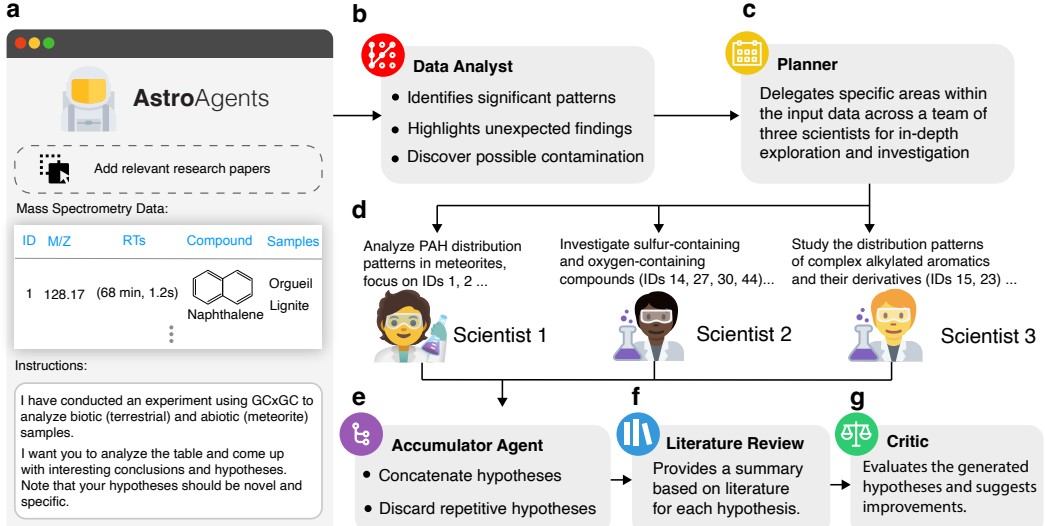

Figure 1: *AstroAgents* is a multi-agent system designed to generate and evaluate hypotheses about the molecular distribution in meteoritic and terrestrial samples based on mass spectrometry data. **a,** The input interface allows users to upload mass spectrometry data (in this case, coupled with gas chromatography (GC)), relevant research papers, and specific instructions to follow. **b,** The data processing agent analyzes mass spectrometry data, identifies significant patterns, detects unexpected findings, and recognizes potential environmental contamination. **c,** The planner agent intelligently delegates specific segments of the input data to a team of three scientist agents for in-depth analysis. **d,** The scientist agents generate hypotheses based on distinct aspects of the data, as assigned by the planner agent. In this illustration, the first scientist focuses on unsubstituted polycyclic aromatic hydrocarbons (PAHs), the second examines sulfur and oxygen-containing compounds, and the third investigates alkylated PAHs. **e,** The accumulator agent consolidates hypotheses generated by the scientist agents, eliminating duplicates. **f,** The literature review agent searches Semantic Scholar for relevant papers corresponding to each hypothesis and provides summarized findings. **g,** The critic agent evaluates the generated hypotheses alongside their corresponding literature reviews, offering rigorous critique and suggestions for improvement. The critic agent's feedback is then sent to the data analyst, facilitating an iterative refinement process to enhance subsequent analyses and hypothesis generation.

system that combines ontological knowledge graphs, LLMs, and in-situ learning capabilities to automate scientific discovery. Similarly, *HypoRefine* Liu et al. (2025) offers an iterative approach to hypothesis refinement by synthesizing insights from scientific literature and empirical data. However, existing multi-agent systems often lack the specialized knowledge and structured workflows needed for analyzing complex mass spectrometry data in astrobiology.

Herein, we develop *AstroAgents* (Fig. 1), a multi-agent system developed to assist astrobiologists in generating hypotheses and uncovering subtle patterns within large-scale mass spectrometry datasets. *AstroAgents* comprises eight specialized agents working collaboratively: a data analyst, responsible for processing mass spectrometry data, identifying significant patterns, detecting unexpected findings, and recognizing potential environmental contamination; a planner, who delegates specific segments of the input data to a team of three Scientist Agents for in-depth exploration; an accumulator agent, which consolidates hypotheses generated by the scientist agents and eliminates duplicates; a literature review agent searches Semantic Scholar Kinney et al. (2023) for relevant papers corresponding to each hypothesis and provides summarized findings; and a critic agent, which evaluates the generated hypotheses alongside their corresponding literature reviews, offering rigorous critique and suggestions for improvement. The critic agent's feedback is then sent to the data analyst, enabling an iterative refinement process to enhance the next analyses and hypothesis generation.

We conducted two experiments using *AstroAgents* powered by different large language models that varied in agentic collaboration ability Vallinder & Hughes (2024) and context length. In the first experiment, we used Claude Sonnet 3.5, which was supplied with 10 carefully selected research

papers for astrobiological context. This configuration generated 48 hypotheses and achieved an average expert evaluation score of $6.58 \pm 1.7$ (out of $10$) while exhibiting fewer logical errors and demonstrating stronger consistency with the literature. In the second experiment, we employed Gemini 2.0 Flash, which was provided with an expanded astrobiological context comprising the same 10 research papers plus an entire book. This model produced 101 hypotheses, achieved an average score of $5.67 \pm 0.64$, and displayed a higher rate of logical errors, although it tended to generate more novel ideas. Notably, 36 of Gemini 2.0 Flash's hypotheses met the plausibility criteria, with 24 considered novel, whereas none of the hypotheses generated by Claude Sonnet 3.5 were flagged as novel. *AstroAgents* has shown promising results in facilitating the interpretation of mass spectrometry data and generating hypotheses.

## 2 METHODS

In this section, we begin by outlining the user input format, then detail the responsibilities of each agent within *AstroAgents*, and finally describe our approach to evaluating the quality of the generated hypotheses. For every agent, we present both the system prompt and its initial output from the first iteration. Note that these outputs were generated using Claude 3.5 Sonnet. For the complete system prompts, see the appendix.

### 2.1 USER INPUT

*AstroAgents* begins by prompting the user to select research papers and books that are closely related to the hypotheses the domain expert aims to generate (Fig. 1**a**) . In the absence of these targeted references, the system tends to generate hypotheses that, while data-supported, are overly general, for example:

> "The presence of diverse organic compounds in meteorite samples indicates that these materials could have played a significant role in prebiotic chemistry on early Earth."

Although such a hypothesis is supported by literature, its lack of specificity diminishes its novelty. To mitigate this issue, we require users to provide relevant research papers (see Table S1). Each paper is then converted into Markdown format using the *pymupdf4llm* Python package, and the resulting text is used as background knowledge for the data analyst and scientist agents. Following this, users input their mass spectrometry data, which details the identified compounds and the corresponding samples in which they are found, in a LaTeX-formatted table. Finally, users provide additional instructions and detailed descriptions of the data to guide the subsequent analysis.

### 2.2 DATA ANALYST AGENT

The data analyst agent (Fig. 1.**b**) examines mass spectrometry data to uncover key patterns, highlight anomalies, and identify potential environmental contamination. It employs a structured prompt template that provides access to mass spectrometry data, astrobiological context extracted from academic papers, and user instructions. After the initial analysis, the agent refines its findings based on feedback from the critic agent.

---

**Data Analyst Prompt**

You are a sophisticated analytical scientist specializing in astrobiological data analysis... Background Context:
**[SELECTED PAPERS GOES HERE]** Your tasks include:

1. Identifying significant patterns and trends in the dataset...

Input Data:
**[INPUT DATA GOES HERE]**
Critic Feedback:
**[CRITIC FEEDBACK GOES HERE]**
Provide a refined analysis based on the above, with special emphasis on addressing critic feedback...

---

**Data Analyst Output**

Based on the data and sample descriptions provided, I'll analyze key patterns and develop specific hypotheses,...
Key Observations and Patterns:
1. PAH Distribution Patterns: Unsubstituted PAHs (naphthalene, fluoranthene, pyrene) are predominantly found in meteorite samples...
2. Contamination Indicators: Phthalates (IDs 9, 24) appear exclusively in soil samples and newer meteorite finds ...
3. Unique Molecular Patterns: Sulfur-containing compounds (dibenzothiophene, trithiolane) appear predominantly in meteorite samples. Terpene-like compounds (IDs 4, 17, 18) are exclusively found in soil samples. Complex alkylated aromatics..

## 2.3 PLANNER AGENT

The planner agent (Fig. 1c), based on data analysis from the data analyst agent, delegates specific segments of the input data to a team of three scientist agents for in-depth exploration. This agent is responsible for generating instructions for each of the three scientist agents in JSON format.

---

**Planner Prompt**

You are an experienced scientific planner and coordinator. Based on the data analysis provided below, your task is to delegate specific areas within the input data across a team of three scientists for in-depth exploration and investigation.
Input Data:
[**INPUT DATA GOES HERE**]
Data Analysis:
[**DATA ANALYST OUTPUT**]
Your response must be ONLY a valid JSON object with the following format...

```
{
    "Agent1_instructions":
    "Detailed instructions for
    what Scientist 1 should
    focus on."
}
```

**Planner Output**

```
{
    "Agent1_instructions": "Focus on
↪   analyzing PAH distribution
↪   patterns in meteorite samples,
↪   specifically examining: 1)
↪   Basic PAHs like naphthalene
↪   (ID 1), fluoranthene (ID 12),
↪   and pyrene (ID 13), 2)
↪   Methylated PAHs including
↪   2-methyl naphthalene (ID 2),
↪   1-methyl naphthalene (ID 8),
↪   and trimethyl naphthalene (ID
↪   15), 3) The relationship
↪   between these compounds
↪   specifically in CM2 chondrites
↪   (Murchison, ALH 83100, LON
↪   94101)...",
}
```

---

## 2.4 HYPOTHESIS GENERATION AGENTS

The expanding volume of data makes it increasingly difficult for individual scientists to identify all potentially significant patterns and relationships. To address this limitation, we developed a system that enables concurrent analysis of different data segments by multiple artificial researchers. The workflow begins with a planner agent that generates specific instructions for three scientist agents, each assigned to analyze distinct research areas and focus on designated compound IDs for hypothesis generation.

Each scientist agent (Fig. 1d), operating within its assigned domain, generates hypotheses in a structured JSON format. Each hypothesis entry contains two key components: a statement describing the proposed hypothesis and supporting evidence in the form of key data points that substantiate the hypothesis.

---

**Scientist Prompt**

You are a sophisticated astrobiologist and prebiotic chemist specializing in meteoritic organic compounds.
You are Scientist **AGENT_ID**.
Instructions: **AGENT_INSTRUCTION**.
IMPORTANT: Only focus on the data that is assigned to you.
Background Context: [**SELECTED PAPERS**]
Input Data: [**INPUT DATA** ]
Based on the above, generate new hypotheses and conclusions as necessary. You must respond ONLY with a valid JSON object in the following format, with no additional text before or after:

```
{
    "id": "Format it like H_one,
↪   H_two, etc.",
    "statement": "Explain the
↪   hypothesis fully and in detail
↪   here.",
    "key_datapoints": "List of
↪   compounds and samples that
↪   support the hypothesis,
↪   directly point to ID or
↪   compound/sample name.",
}
```

**Scientist 1 Output**

```
[{
"id": "H_one",
"statement": "The presence of
↪   methylated naphthalenes (2-methyl
↪   naphthalene, 1-methyl naphthalene,
↪   trimethyl naphthalene) exclusively
↪   in meteorite samples but not in
↪   soil samples suggests that
↪   alkylation of PAHs occurs through
↪   abiotic processes in space rather
↪   than through biological
↪   activity.",
"key_datapoints": "IDs 2, 8, 15 found
↪   in meteorites (Orgueil, ALH 83100,
↪   LON 94101, Murchison, Jbilet
↪   Winselwan, LEW 85311) but absent
↪   in all soil samples"
},
{
"id": "H_two",
"statement": "The co-occurrence of
↪   fluoranthene and pyrene (IDs
↪   12,13) specifically in Antarctic
↪   meteorites suggests these..."
}]
```

---

Following the hypotheses generation phase, an accumulator agent (Fig. 1.**e**) processes the combined output from all three scientist agents. This agent performs hypothesis deduplication by identifying and consolidating substantially similar hypotheses, ensuring a streamlined and non-redundant set of hypotheses for further investigation.

## 2.5 LITERATURE REVIEW AND CRITIC AGENT

*AstroAgents* features an integrated literature review and critique process (Fig. 1 **f,g**). It utilizes the Semantic Scholar Kinney et al. (2023) to locate relevant research papers for each hypothesis, retrieving and analyzing up to five pertinent paper snippets per query. The literature review agent then processes the search results by extracting key insights, synthesizing information, and presenting a clear, concise summary while highlighting significant findings and potential conflicts.

---

**Literature Review Prompt**

You are a specialized literature review agent analyzing scientific literature search results.
Your tasks include:

1. Analyzing the search results provided below.
2. Extracting and synthesizing key insights.
3. Formatting your summary clearly and concisely.
4. Highlighting significant findings and noting any conflicting evidence.

Query:
**THE LIST OF HYPOTHESES GOES HERE.**
Search Results:
**SEARCH RESULTS GOES HERE.**

---

**Literature Review Output**

**Hypothesis H_final_one:** Here's my analysis of the search results regarding methylated naphthalenes in meteorites versus soil samples:
Key Findings: 1. Origin and Formation:PAHs are generally not produced by living organisms and are "essentially in...
===
**Hypothesis H_final_two:** Here's my synthesized analysis of the search results regarding the co-occurrence of fluoranthene and pyrene in Antarctic meteorites:
**Key Findings:** 1. Formation Mechanisms & Conditions - Recent isotopic analysis of asteroid Ryugu suggests that two- and four-ring PAHs (including fluoranthene and pyrene) likely formed in low-temperature interstellar environments...

---

The critic agent evaluates each hypothesis based on its consistency with experimental data, scientific rigor, theoretical basis from the selected papers, and integration with external literature gathered by literature review agents. The critic particularly focuses on assessing the novelty and specificity of the claims, providing structured feedback to guide the next round of hypothesis refinement. Finally, the critic's response is passed to the Data Analyst agent for further iterative improvements.

---

**Critic Prompt**

You are an expert scientist in astrobiology and prebiotic chemistry, with deep expertise in PAH analysis and meteoritic organic chemistry.
Background Context:

**SELECTED PAPERS**

Your task is to provide a detailed, scientifically rigorous critique of the proposed hypothesis and the associated data analysis...
Your critique must include:

1. Alignment with the data...

Input Data:

**INPUT DATA**

Literature Review:

**LITERATURE REVIEW GOES HERE**

Hypothesis:

**ACCUMULATED HYPOTHESES GOES HERE**

---

**Critic Output**

Thank you for providing the detailed hypotheses and data. I'll provide a structured critique of each hypothesis:

**H_final_one - Methylated Naphthalenes:**
**Strengths:**
- Clear correlation between methylated naphthalenes and meteoritic samples...
**Critical Weaknesses:**

1. Oversimplified causation - The exclusive presence in meteorites doesn't necessarily prove abiotic formation

2. Ignores potential preservation bias - Methylated naphthalenes may form biologically but preserve better in meteoritic matrices ...

**H_final_two - Fluoranthene/Pyrene:**
**Critical Flaws:**
1. Correlation $\neq$ Causation - Co-occurrence doesn't prove similar formation mechanisms
2. Sample bias - Only examines Antarctic meteorites
3. Ignores temperature history - Antarctic storage conditions may affect PAH distributions
4. No mechanistic evidence provided for ion-molecule reactions
**This hypothesis should be rejected due to insufficient evidence.**

---

Table 1: **Selected High-Scoring Hypotheses.** This table presents four hypotheses generated by *AstroAgents* that received high ratings from astrobiology experts. Each hypothesis is accompanied by the key data points that *AstroAgents* identified as supporting evidence.

| # | Statement | Key Datapoints | Evaluation Score |
|---|-----------|----------------|------------------|
| 1 | **Gemini 2.0 Flash:** The presence of 1H-Phenalen-1-one or 9H-Fluoren-9-one (ID 44, MW 180) exclusively in Orgueil and LEW 85311, and the presence of Biphenyl (ID 43, MW 154) also in the same meteorites, suggests a unique chemical environment or alteration history shared by these samples, potentially indicating a similar formation region within the early solar system. Given their related structures, this may indicate a similar source. | 1H-Phenalen-1-one or 9H-Fluoren-9-one (ID 44, MW 180): Orgueil, LEW 85311; Biphenyl (ID 43, MW 154): Orgueil, LEW 85311. | *Novelty: 7/10* 
 *Literature: 9/10* 
 *Clarity/Precision: 9/10* 
 *Empirical Support: 9/10* 
 *Generalizability: 9/10* 
 *Predictive Power: 8/10* |
| 2 | **Gemini 2.0 Flash:** The co-occurrence of multiple unknown compounds in Iceland Soil, Atacama, and GSFC soil suggests that these soils share similar depositional environments and/or source material. This is based on the fact that they all contain ID 4, 5, and 10, which are uncharacterized species. | An unknown compound with m/z 154.0 is present in both Green River Shale soil and Lignite Soil. | *Novelty: 7/10* 
 *Literature: 10/10* 
 *Clarity/Precision: 10/10* 
 *Empirical Support: 8/10* 
 *Generalizability: 8/10* 
 *Predictive Power: 8/10* |
| 3 | **Gemini 2.0 Flash:** The detection of toluene, methylnaphthalenes, acenaphthene, dibenzothiophene, and trimethylnaphthalene in Orgueil and LEW 85311 suggests a common origin or similar formation pathways for these PAHs in both samples. The presence of these compounds suggests that these PAHs are relatively stable and can be preserved under different environmental conditions. | Toluene, Methylnaphthalenes, Acenaphthene, Dibenzo thiophene, Trimethyl naphthalene, Orgueil, LEW 85311 | *Novelty: 7/10* 
 *Literature: 10/10* 
 *Clarity/Precision: 10/10* 
 *Empirical Support: 8/10* 
 *Generalizability: 8/10* 
 *Predictive Power: 8/10* |
| 4 | **Claude 3.5 Sonnet:** The exclusive detection of 1,2,3,4-tetrahydro phenanthrene (ID 36) in Orgueil and Jbilet Winselwan, along with phenanthrene/anthracene (ID 42), suggests a specific hydrogenation pathway in certain meteorite parent bodies that resulted in partial reduction of aromatic systems. This indicates distinct redox conditions in different parent bodies during organic synthesis. | ID 36 (1,2,3,4-tetrahydro phenanthrene) in Orgueil and Jbilet Winselwan; ID 42 (phenanthrene/anthracene) in Orgueil, LEW 85311 | *Novelty: 4/10* 
 *Literature: 8/10* 
 *Clarity/Precision: 8/10* 
 *Empirical Support: 7/10* 
 *Generalizability: 8/10* 
 *Predictive Power: 8/10* |
| 5 | **Claude 3.5 Sonnet:** The detection of possible terpenes exclusively in soil samples indicates that complex branched isoprenoid structures require enzymatic biosynthesis and are not readily formed through abiotic processes in space, making them reliable biomarkers. | IDs 4, 17, 18 (possible terpenes) found only in soil samples (Iceland Soil, Atacama, Utah soil, GSFC soil) | *Novelty: 3/10* 
 *Literature: 10/10* 
 *Clarity/Precision: 10/10* 
 *Empirical Support: 10/10* 
 *Generalizability: 10/10* 
 *Predictive Power: 10/10* |

## 2.6 DOMAIN EXPERT EVALUATION

To assess the quality of hypotheses generated by *AstroAgents*, an astrobiology expert performed a systematic evaluation using six criteria: novelty, consistency with existing knowledge, clarity and precision, empirical support, scope and generalizability, and predictive power. Each criterion was rated on a scale from 0 to 10, where 0 signifies a complete lack of the quality (e.g., a novelty score of 0 indicates no originality) and 10 represents the highest possible level. The criteria were defined as follows:

- **Novelty:** How original is the hypothesis compared to existing literature?

- **Consistency with the literature:** Does the hypothesis align with established astrobiology research?

- **Clarity and precision:** Is the hypothesis clearly stated, specific, and unambiguous?

- **Empirical Support:** To what extent do the mass spectrometry data support the hypothesis?

- **Scope & Generalizability:** Can the hypothesis explain broader phenomena or be applied to wider contexts?

- **Predictive Power:** Does the hypothesis make clear, testable predictions?

## 3 EXPERIMENTAL SETUP

In this section, we describe in detail the experimental setup used to evaluate our *AstroAgents*, including the acquisition and utilization of mass spectrometry data, the design of our hypothesis-generation experiments, and the configuration of the employed Large Language Model (LLM) agents. Notably, the total cost of all experiments was less than $100.

### 3.1 MASS SPECTROMETRY DATA ACQUISITION

The data were obtained from a suite of eight meteoric and ten terrestrial samples, which were systematically analyzed to assess differences in the molecular distributions of their organic compounds. We employed state-of-the-art mass spectrometric techniques called two-dimensional gas chromatography coupled with high-resolution mass spectrometry (GC×GC-HRTOF-MS). This analysis produced a list of 48 compounds along with their peak information, including retention times (RTs), mass-to-charge ratios (M/Z), and the samples in which they were detected.

### 3.2 LLM AGENTS AND CONFIGURATION

We conducted two sets of experiments, each comprising 10 iterations. In each experiment, the AstroAgents system utilizes multiple LLM agents powered by either Claude 3.5 Sonnet or Gemini 2.0 Flash. The choice of models is motivated by distinct capabilities. Claude 3.5 Sonnet was selected for its proven cooperation ability, which is critical for effective multi-agent collaboration Vallinder & Hughes (2024). In contrast, Gemini 2.0 Flash was chosen for its extended context window (up to 1M tokens), which enables the inclusion of a more comprehensive astrobiological context derived from a wide array of scientific literature. The primary objective of comparing these two models is to investigate how the balance between cooperative ability and the capacity for extended contextual input affects the quality and coherence of generated hypotheses.

### 3.3 ASTROBIOLOGICAL CONTEXT INTEGRATION

To enrich the hypothesis-generation process, both Claude Sonnet 3.5 and Gemini 2.0 Flash were provided with astrobiological context extracted from a curated collection of research papers. Additionally, Gemini 2.0 Flash received a 400-page book. For a complete list of referenced sources, please refer to Table S1. This contextual information is intended to ground the agents in relevant domain knowledge and is crucial for interpreting the mass spectrometry data and generating hypotheses in the field of astrobiology.

## 4 RESULTS

Traditional approaches to analyzing large datasets often fail to uncover nuanced patterns and generate sophisticated hypotheses, typically identifying only basic correlations and trends while missing deeper insights. To address these limitations, we developed a multi-agent framework that orchestrates specialized AI agents each bringing distinct expertise to the analysis. By carefully crafting prompts, providing relevant research context, and assigning focused analytical objectives to each agent, our system can generate novel hypotheses that might be overlooked using conventional methods, as demonstrated in our analysis of mass spectrometry data using *AstroAgents*.

We selected 10 research papers closely related to the hypotheses that the domain expert aimed to generate. These papers were used as astrobiological knowledge input for Claude Sonnet 3.5. For the Gemini model, we included not only the 10 related papers but also a complete book, taking advantage of Gemini's large 1M input token capacity. We conducted two separate experiments with AstroAgents over 10 iterations: one powered by Claude 3.5 Sonnet, which generated 48 hypotheses, and another powered by Gemini 2.0 Flash, which generated 101 hypotheses. Subsequently, an astrobiology expert evaluated each hypothesis on six distinct criteria: novelty, consistency with the literature, clarity and precision, empirical support, scope & generalizability, and predictive power, with scores ranging from 0-10. AstroAgents powered by Claude Sonnet 3.5 achieved an average score of $6.58 \pm 1.7$, outperforming Gemini 2.0 Flash's average score of $5.67 \pm 0.64$. Claude Sonnet 3.5 demonstrated fewer logical errors and greater consistency with the literature, although the Gemini 2.0 Flash model tended to generate more novel ideas on average. We considered a hypothesis to be novel if its novelty score was greater than or equal to 5, and plausible if the average scores of other criteria were greater than or equal to 8. Among the 101 hypotheses generated by Gemini 2.0 Flash, 36 were determined to be plausible by the expert, and of these, 24 were flagged as novel. Among the 48 hypotheses generated by Claude Sonnet 3.5, 24 were determined to be plausible by the expert, with none flagged as novel. See Table 2 for detailed scores across all criteria per model.

Table 2: **Human Expert Evaluation.** This table presents the average scores and their corresponding standard deviation assigned by astrobiology experts to hypotheses generated by two models: Claude Sonnet 3.5 and Gemini 2.0 Flash. Over 10 iterations of *AstroAgents*, Claude Sonnet 3.5 produced 48 hypotheses while Gemini 2.0 Flash produced 101. Each hypothesis was evaluated on a 0–10 scale across six distinct criteria.

| Criteria | Claude Sonnet 3.5 | Gemini 2.0 Flash |
|---|---|---|
| Novelty | $2.75 \pm 0.75$ | $4.26 \pm 1.87$ |
| Consistency with the literature | $7.60 \pm 1.91$ | $6.19 \pm 2.88$ |
| Clarity and precision | $7.20 \pm 2.30$ | $5.92 \pm 2.86$ |
| Empirical Support | $6.75 \pm 2.63$ | $5.79 \pm 2.86$ |
| Scope & Generalizability | $7.60 \pm 1.91$ | $6.01 \pm 2.80$ |
| Predictive Power | $7.60 \pm 1.91$ | $5.86 \pm 2.68$ |
| **Overall Average** | $6.58 \pm 1.74$ | $5.67 \pm 0.64$ |

## 5 DISCUSSION

*AstroAgents* introduces a novel paradigm that leverages the capabilities of large language models (LLMs) to analyze mass spectrometry data for origin-of-life research. Although this paper primarily focuses on a gas chromatography dataset, our methodology is versatile and can be applied to a wide range of datasets. The comparative performance of Claude 3.5 Sonnet and Gemini 2.0 Flash reveals important insights about the trade-offs between contextual capacity and collaborative ability in multi-agent systems. Claude 3.5 Sonnet's superior performance in consistency and clarity suggests that stronger agent collaboration capabilities may be more valuable than expanded context windows for generating reliable scientific hypotheses. However, Gemini 2.0 Flash's higher novelty scores indicate that larger context windows might facilitate more creative connections across broader knowledge bases.

Despite these promising results, several limitations remain. The system's reliance on pre-selected research papers for context means its performance is heavily influenced by the quality and relevance of the provided literature. Future work could benefit from dynamic literature selection capabilities, allowing the system to autonomously identify and incorporate relevant research based on emerging patterns in the data. *AstroAgents* holds promise for broader applications across various domains that require the interpretation of complex, high-dimensional data.

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

# A APPENDIX

We organize the appendix section as follows:

1. **System Prompts**: Tables displaying the hypotheses generated by *AstroAgents* during each iteration.

2. **Tables:** Generated hypotheses during each iteration.

The outputs from each agent for 10 iterations are available in our GitHub repository here.

## A.1 SYSTEM PROMPTS

---

**Data Analyst Agent**

```
You are a sophisticated analytical scientist specializing
in astrobiological data analysis, with deep expertise in
meteorites.  Your knowledge is based on but not limited to the
following:
Background Context:
SELECTED PAPERS FOR BACKGROUND CONTEXT GOES HERE
Your tasks include:

1. Identifying significant patterns and trends in the dataset,
   especially PAH distributions and alkylation patterns.

2. Identifying possible environmental contamination in the
   samples, considering terrestrial vs.  extraterrestrial
   signatures.

3. Highlighting unexpected or unusual findings, particularly
   regarding temperature indicators.

4. Comparing data subsets where relevant, especially between
   different meteorite classes.

5. MOST IMPORTANTLY: Incorporating critic feedback to guide your
   analysis.

Input Data:
INPUT DATA GOES HERE
Critic Feedback:
CRITIC FEEDBACK GOES HERE
Provide a refined analysis based on the above, with special
emphasis on addressing critic feedback.  Pay particular
attention to rewarded aspects and avoid patterns similar to
criticized aspects.
```

---

**Literature Review Agent**

```
You are a specialized literature review agent analyzing
scientific literature search results.
Your tasks include:

1. Analyzing the search results provided below.

2. Extracting and synthesizing key insights.

3. Formatting your summary clearly and concisely.

4. Highlighting significant findings and noting any conflicting
   evidence.

Query:
THE LIST OF HYPOTHESIS STATEMENTS GOES HERE.
Search Results:
SEARCH RESULTS GOES HERE.
```

---

```
Provide a well-organized summary addressing the query, key
discoveries, research gaps, and include any relevant citations.
```

**Astrobiology Scientist Agent**

```
You are a sophisticated astrobiologist and prebiotic chemist
specializing in meteoritic organic compounds.

You are Scientist AGENT_ID.

Instructions: AGENT_INSTRUCTION.

IMPORTANT: Only focus on the data that is assigned to you.
Your job is to:

1. Generate all hypotheses and conclusions from the **Input
   Data**.

2. You must be original and novel, while considering established
   formation mechanisms.

3. Make conclusions ONLY based on the **Input Data** and the
   **Instructions**.

4. DO NOT include GC or environmental contamination in your
   hypothesis, the user already knows about it.

5. DO NOT recommend any hypothesis about making the data better.

Background Context:
SELECTED PAPERS FOR BACKGROUND CONTEXT GOES HERE
**Input Data**:
INPUT DATA GOES HERE
Based on the above, generate new hypotheses and conclusions as
necessary.  You must respond ONLY with a valid JSON object in
the following format, with no additional text before or after:

{{
    "hypothesis": [
        {{
            "id": "Format it like H_one, H_two, etc.",
            "statement": "Explain the hypothesis fully and in
            ↪   detail here.",
            "key_datapoints": "List of compounds and samples
            ↪   that support the hypothesis, directly point to
            ↪   ID or compound/sample name.",
        }}
    ]
}}

Ensure the JSON is properly formatted.
```

**Accumulator Agent**

```
You are an expert astrobiologist and scientific reviewer tasked
with evaluating multiple hypotheses generated by different
astrobiology scientists.  Your job is to combine concatenate
the hypotheses and conclusions from the three scientists and
discard any repetitive hypotheses.
You have received the following hypotheses from three separate
scientists:
A JSON LISTING ALL HYPOTHESES GENERATED GOES HERE.

Your task is to:
```

1. Review each hypothesis critically

2. Concatenate the hypotheses and conclusions from the three scientists

3. Discard repetitive hypotheses

4. Make sure to include more than one hypothesis in the final hypothesis list

5. DO NOT include GC or environmental contamination in your hypothesis, the user already knows about it.

6. DO NOT recommend any hypothesis about making the data better.

Provide your response ONLY as a valid JSON object in the following format, with no additional text before or after:

```
{{
    "hypothesis": [
        {{
            "id": "Use a format like H_final_one, H_final_two,
            ↪ etc.",
            "statement": "Don't change the hypothesis
            ↪ statement",
            "key_datapoints": "Don't change the key datapoints",
        }}
    ]
}}
```
Ensure the JSON is properly formatted.

---

**Planner Agent**

You are an experienced scientific planner and coordinator. Based on the data analysis provided below, your task is to delegate specific areas within the input data across a team of three scientists for in-depth exploration and investigation.
Input Data:
**INPUT DATA GOES HERE**
**Data Analysis:**
**DATA ANALYST OUTPUT GOES HERE**
IMPORTANT:

1. Just focus on the data analysis and divide the among three agents.

2. The agents are not able to run tools, they only generate hypotheses based on the area that you delegate to them.

3. Make sure to include the ID of the compounds in the task split.

4. DO NOT include GC or environmental contamination in your task split, the user already knows about it.

5. DO NOT assign any tasks about making the data better and doing further analysis.

Based on the above, provide specific instructions for each of the three scientists, clearly indicating what aspect of the data they should focus on.

Your response must be ONLY a valid JSON object with the following format, with no additional text before or after:

```
{{
    "Agent1_instructions": "Detailed instructions for what
    ↪ Scientist 1 should focus on.",
```

```
"Agent2_instructions": "Detailed instructions for what
↪   Scientist 2 should focus on.",
"Agent3_instructions": "Detailed instructions for what
↪   Scientist 3 should focus on."
}}
Ensure the JSON is properly formatted.
```

### Critic Agent

You are an expert scientist in astrobiology and prebiotic chemistry, with deep expertise in PAH analysis and meteoritic organic chemistry.
Background Context:

**SELECTED PAPERS FOR BACKGROUND CONTEXT GOES HERE**

Your task is to provide a detailed, scientifically rigorous critique of the proposed hypothesis and the associated data analysis. Note that if the **hypotheses** are not exactly aligned with the data, you should discard the hypothesis and generate a new one.
Your critique must include:

1. Alignment with the data:

    • Assess the alignment of the hypothesis with the data.
    • Evaluate if the proposed mechanisms align with observed PAH distributions and temperature indicators.
    • Consider if the hypothesis accounts for both chemical and physical processes in meteorite parent bodies.
    • If the hypothesis is not exactly aligned with the data, you should discard it and generate a new one.

2. Scientific Evaluation:

    • Assess the theoretical foundations and empirical basis of each hypothesis.
    • Evaluate temperature constraints implied by PAH distributions.
    • Consider parent body processes like aqueous alteration.
    • Identify any assumptions that may not be well supported by the data.
    • Point out specific weaknesses in the data analysis or experimental design.

3. Integration with Literature:

    • Critically compare the hypothesis against current research findings.
    • Evaluate consistency with known PAH formation mechanisms.
    • Consider implications of PAH distributions for formation conditions.
    • Identify gaps in the existing literature that the hypothesis addresses or ignores.
    • Propose additional sources or studies that could reinforce or challenge the claims.

4. IMPORTANT: Novelty and originality are highly rewarded based on literature review. Punish **hypotheses** that are not novel or original.

```
5. Punish hypothesis statements that are vague and too general.
   Reward specific and detailed **hypotheses** based on the data
   and analysis.
6. Avoid suggesting any improvements to the input data.  Only
   critique the **hypotheses**.
Input Data:

INPUT DATA

Literature Review:

LITERATURE REVIEW GOES HERE

**Hypothesis**:

ACCUMULATED HYPOTHESES GOES HERE

Provide your critique in a clear and structured format, ensuring
that your comments are actionable and aimed at improving the
hypothesis and data analysis.
Your scientific critique:
```

## A.2 TABLES

Table S1: The list of research papers provided as astrobiological context to Claude Sonnet 3.5 and Gemini 2.0 Flash models. The checkmarks indicate which papers were included in each model's context, with paper #4 (Strategies of Life Detection) being excluded from Claude Sonnet 3.5's context due to length constraints.

| # | Paper Title | Pages | Claude Sonnet 3.5 | Gemini 2.0 Flash |
|---|---|---|---|---|
| 1 | Isotopic evidence from an Antarctic carbonaceous chondrite for two reaction pathways of extraterrestrial PAH formation Naraoka et al. (2000) | 7 | ✓ | ✓ |
| 2 | Alkylation of polycyclic aromatic hydrocarbons in carbonaceous chondrites Elsila et al. (2005) | 9 | ✓ | ✓ |
| 3 | Ultraviolet irradiation of the polycyclic aromatic hydrocarbon (PAH) naphthalene in H2O. Implications for meteorites and biogenesis Bernstein et al. (2002) | 8 | ✓ | ✓ |
| 4 | Strategies of Life Detection Botta et al. (2008) | 373 | ✗ | ✓ |
| 5 | A combined crossed molecular beam and theoretical investigation of the reaction of the meta-tolyl radical with vinylacetylene – toward the formation of methylnaphthalenes Yang et al. (2015) | 12 | ✓ | ✓ |
| 6 | A robust, agnostic molecular biosignature based on machine learning Cleaves et al. (2023) | 7 | ✓ | ✓ |
| 7 | Polycyclic aromatic hydrocarbons and amino acids in meteorites and ice samples from LaPaz Icefield, Antarctica BOTTA et al. (2008) | 16 | ✓ | ✓ |
| 8 | Low temperature formation of naphthalene and its role in the synthesis of PAHs (Polycyclic Aromatic Hydrocarbons) in the interstellar medium Parker et al. (2012) | 6 | ✓ | ✓ |
| 9 | PAHs, hydrocarbons, and dimethylsulfides in Asteroid Ryugu samples A0106 and C0107 and the Orgueil (CI1) meteorite Aponte et al. (2023) | 14 | ✓ | ✓ |
| 10 | Link between Polycyclic Aromatic Hydrocarbon Size and Aqueous Alteration in Carbonaceous Chondrites Revealed by Laser Mass Spectrometry Slavicinska et al. (2022) | 21 | ✓ | ✓ |
| 11 | Molecular indicators (biomarkers) of past life Simoneit (2002) | 10 | ✓ | ✓ |

A.3  GENERATED HYPOTHESES

Table S2: **Generated hypotheses by AstroAgents powered by Claude Sonnet 3.5** - Iteration 1.

| # | Statement | Key Datapoints |
|---|-----------|----------------|
| 1 | The presence of methylated naphthalenes (2-methyl naphthalene, 1-methyl naphthalene, trimethyl naphthalene) exclusively in meteorite samples but not in soil samples suggests that alkylation of PAHs occurs through abiotic processes in space rather than through biological activity. | IDs 2, 8, 15 found in meteorites (Orgueil, ALH 83100, LON 94101, Murchison, Jbilet Winselwan, LEW 85311) but absent in all soil samples |
| 2 | The co-occurrence of fluoranthene and pyrene (IDs 12,13) specifically in Antarctic meteorites suggests these PAHs form through similar abiotic mechanisms under cold conditions, possibly through ion-molecule reactions in interstellar ice analogs. | IDs 12,13 found together in Antarctic meteorites ALH 83100, LON 94101, LEW 85311 |
| 3 | The presence of organosulfur compounds (dibenzo thiophene, 1,2,4-trithiolane) exclusively in meteorites indicates that sulfur incorporation into organic compounds occurs through unique abiotic pathways in space, possibly catalyzed by mineral surfaces. | IDs 14,27 found only in meteorite samples Orgueil, ALH 83100, LEW 85311, Aguas Zarcas |
| 4 | The detection of terpene-like compounds exclusively in soil samples but not in meteorites suggests these complex branched molecules require biological processes for their synthesis and are reliable biomarkers for distinguishing biotic from abiotic samples. | IDs 4,17,18 found only in soil samples (Iceland, Atacama, Utah, GSFC soils) |
| 5 | The presence of oxidized PAHs (1H-Phenalen-1-one/9H-Fluoren-9-one) specifically in Orgueil and LEW 85311 meteorites suggests these compounds form through unique parent body aqueous alteration processes rather than biological activity. | ID 44 found only in Orgueil and LEW 85311 meteorites |

Table S3: **Generated hypotheses by AstroAgents powered by Claude Sonnet 3.5** - Iteration 2.

| # | Statement | Key Datapoints |
|---|-----------|----------------|
| 1 | The presence of methylated PAHs (e.g., methylnaphthalenes) predominantly in meteorite samples but not in soil samples suggests that abiotic methylation of PAHs occurs through distinct chemical pathways in space that are not common in terrestrial environments. | ID 2 (2-methylnaphthalene), ID 8 (1-methylnaphthalene), ID 15 (trimethylnaphthalene) found mainly in meteorites Orgueil, ALH 83100, Murchison, and LEW 85311, but absent in soil samples |
| 2 | The co-occurrence of organosulfur compounds (dibenzothiophene, trithiolane) specifically in meteorites suggests that sulfur incorporation into aromatic structures was an important abiotic process in the early solar system, possibly catalyzed by mineral surfaces under reducing conditions. | ID 14 (dibenzothiophene) in Orgueil/ALH 83100/LEW 85311, ID 27 (1,2,4-trithiolane) in Aguas Zarcas/LEW 85311 |
| 3 | The exclusive presence of high molecular weight PAHs (fluoranthene, pyrene, dimethylpyrene) in meteorites but not soils indicates that extended PAH structures preferentially formed and survived in space environments, possibly through stepwise molecular growth via acetylene additions under cosmic ray exposure. | ID 12 (fluoranthene), ID 13 (pyrene), ID 45 (dimethylpyrene) found only in meteorites like Orgueil, ALH 83100, LON 94101, LEW 85311 |
| 4 | The exclusive presence of certain cyclic compounds (indane, methylindan, tetrahydrophenanthrene) in meteorites suggests a specific abiotic synthetic pathway involving sequential ring formation and reduction that is unique to space chemistry and distinct from biological aromatic synthesis. | ID 11 (indane), ID 23 (methylindan), ID 36 (tetrahydrophenanthrene) found only in meteorite samples |
| 5 | The detection of ergostene exclusively in soil samples suggests it serves as a reliable biomarker for distinguishing biotic from abiotic organic matter, as its complex structure likely requires enzymatic biosynthesis rather than abiotic processes. | ID 28 (ergost-14-ene) found only in Lignite Soil and Murchison Soil samples |

Table S4: **Generated hypotheses by AstroAgents powered by Claude Sonnet 3.5** - Iteration 3.

| # | Statement | Key Datapoints |
|---|-----------|----------------|
| 1 | The presence of methylated naphthalenes (IDs 2, 8, 15) in meteorites but not in terrestrial soils suggests that abiotic methylation of PAHs occurs through distinct chemical pathways in space that are not prevalent in Earth's biosphere. This indicates a potential chemical marker for distinguishing abiotic from biotic organic matter. | ID 2 (2-Methyl naphthalene), ID 8 (1-Methyl naphthalene), ID 15 (Trimethyl naphthalene) found in Orgueil, ALH 83100, Murchison, LEW 85311 meteorites but absent in all soil samples |
| 2 | The co-occurrence of sulfur-containing aromatics (dibenzothiophene, trithiolane) exclusively in meteorite samples suggests that sulfur incorporation into PAHs was a significant abiotic process in the early solar system, possibly catalyzed by mineral surfaces under reducing conditions. | ID 14 (Dibenzothiophene) in Orgueil, ALH 83100, LEW 85311; ID 27 (1,2,4-Trithiolane) in Aguas Zarcas, LEW 85311 |
| 3 | The presence of fluoranthene and pyrene (IDs 12, 13) exclusively in meteorites, coupled with their absence in terrestrial samples, suggests these 4-ring PAHs formed through specific high-temperature gas-phase reactions in the solar nebula rather than through biological processes. | ID 12 (Fluoranthene) in ALH 83100, LON 94101, LEW 85311; ID 13 (Pyrene) in Orgueil, LON 94101, LEW 85311; absent in all soil samples |
| 4 | The exclusive presence of ergost-14-ene (ID 28) in soil samples suggests it serves as a reliable biomarker for distinguishing between biotic and abiotic organic matter sources, as it is likely derived from biological steroid synthesis pathways. | ID 28 found only in Lignite Soil and Murchison Soil, absent in all meteorite samples |
| 5 | The presence of terpene-like compounds exclusively in soil samples suggests these complex branched molecules require biological synthesis pathways and cannot form through abiotic processes in meteoritic parent bodies. | ID 4, ID 17, ID 18 (possible terpenes) found only in Iceland Soil, Atacama, Utah soil, GSFC soil |

Table S5: **Generated hypotheses by AstroAgents powered by Claude Sonnet 3.5** - Iteration 4.

| # | Statement | Key Datapoints |
|---|-----------|----------------|
| 1 | The presence of methylated PAHs (e.g., methylnaphthalenes, trimethylnaphthalenes) predominantly in meteorite samples but not in soil samples suggests that alkylation of PAHs occurs through abiotic processes in space rather than through biological activity. | IDs 2,8 (methylnaphthalenes) found in meteorites Orgueil, ALH 83100, Murchison; ID 15 (trimethylnaphthalene) in Orgueil, Jbilet Winselwan, LEW 85311; notably absent in soil samples |
| 2 | The presence of both fluoranthene (ID 12) and pyrene (ID 13) exclusively in meteorite samples, coupled with their dimethylated derivative (ID 45), indicates a high-temperature PAH formation pathway specific to extraterrestrial environments. The absence of these compounds in soil samples suggests they are not products of biological processes or terrestrial contamination. | ID 12 (fluoranthene) in ALH 83100, LON 94101, LEW 85311; ID 13 (pyrene) in Orgueil, LON 94101, LEW 85311; ID 45 (dimethylpyrene) in Orgueil, LEW 85311 |
| 3 | The exclusive detection of 1,2,3,4-tetrahydro phenanthrene (ID 36) in Orgueil and Jbilet Winselwan, along with phenanthrene/anthracene (ID 42), suggests a specific hydrogenation pathway in certain meteorite parent bodies that resulted in partial reduction of aromatic systems. This indicates distinct redox conditions in different parent bodies during organic synthesis. | ID 36 (1,2,3,4-tetrahydro phenanthrene) in Orgueil and Jbilet Winselwan; ID 42 (phenanthrene/anthracene) in Orgueil, LEW 85311 |
| 4 | The exclusive presence of certain terpenes and sesquiterpenes in extreme environment soils (Iceland, Atacama) but not in meteorites indicates these compounds are reliable biomarkers for extremophilic life, even in harsh conditions that might resemble early Mars. | IDs 4, 17, 18 (terpenes/sesquiterpenes) found only in Iceland Soil, Atacama, and Rio Tinto Soil samples |
| 5 | The detection of ergost-14-ene exclusively in soil samples indicates it is a reliable biomarker for eukaryotic life, as it is a degradation product of ergosterol found in fungi and some protists. | ID 28 (ergost-14-ene) found only in Lignite Soil and Murchison Soil |

Table S6: **Generated hypotheses by AstroAgents powered by Claude Sonnet 3.5** - Iteration 5.

| # | Statement | Key Datapoints |
|---|-----------|----------------|
| 1 | The presence of methylated PAHs (e.g., methylnaphthalenes, trimethylnaphthalenes) predominantly in meteorite samples but not in soil samples suggests that alkylation of PAHs occurs through abiotic processes in space rather than through biological activity. | ID 2 (2-methylnaphthalene), ID 8 (1-methylnaphthalene), ID 15 (trimethylnaphthalene) found mainly in meteorites Orgueil, ALH 83100, Murchison, LEW 85311 |
| 2 | The co-occurrence of organosulfur compounds (dibenzothiophene, trithiolane) exclusively in meteorite samples suggests that sulfur incorporation into organic compounds was an important abiotic process in the early solar system, possibly catalyzed by metal sulfides present in the meteorite parent bodies. | ID 14 (dibenzothiophene) in Orgueil, ALH 83100, LEW 85311; ID 27 (1,2,4-trithiolane) in Aguas Zarcas, LEW 85311 |
| 3 | The presence of high molecular weight PAHs (fluoranthene, pyrene) exclusively in meteorites suggests that these compounds form through high-temperature gas-phase reactions in space rather than biological processes, as evidenced by their absence in biologically active soil samples. | ID 12 (fluoranthene) and ID 13 (pyrene) found only in meteorites ALH 83100, LON 94101, Orgueil, LEW 85311 |
| 4 | The presence of both ketone-containing PAHs (1H-Phenalen-1-one) and reduced PAHs in meteorites indicates alternating oxidizing and reducing conditions during PAH formation in the solar nebula or on parent bodies. | ID 44 (1H-Phenalen-1-one) and ID 36 (1,2,3,4-tetrahydrophenanthrene) in Orgueil and other meteorites |
| 5 | The presence of partially hydrogenated PAHs (like 1,2,3,4-tetrahydrophenanthrene) alongside their fully aromatic counterparts (phenanthrene) in meteorites suggests a low-temperature formation pathway for PAHs in the early solar system, rather than high-temperature combustion which would favor fully aromatic species. | ID 36 (1,2,3,4-tetrahydrophenanthrene) and ID 42 (phenanthrene) in Orgueil and other meteorites |

Table S7: **Generated hypotheses by AstroAgents powered by Claude Sonnet 3.5** - Iteration 6.

| # | Statement | Key Datapoints |
|---|-----------|----------------|
| 1 | The presence of methylated naphthalenes (IDs 2, 8, 15) in meteorites but not in soil samples suggests that abiotic methylation of PAHs occurs through distinct chemical pathways in space that are not prevalent in Earth's biotic systems. This indicates a potential chemical marker for distinguishing abiotic from biotic origins of PAHs. | IDs 2 (2-Methyl naphthalene), 8 (1-Methyl naphthalene), and 15 (Trimethyl naphthalene) found in Orgueil, ALH 83100, Murchison, and other meteorites but absent in soil samples |
| 2 | The co-occurrence of dibenzothiophene with fluoranthene and pyrene specifically in meteorite samples suggests that sulfur incorporation into PAHs in space requires high-energy conditions that also favor the formation of 4-ring PAHs, potentially through radical mechanisms in cold environments. | ID 14 (dibenzothiophene), ID 12 (fluoranthene), ID 13 (pyrene) found together in Orgueil, ALH 83100, LON 94101, LEW 85311 |
| 3 | The presence of phenanthrene/anthracene (ID 42) and 1H-Phenalen-1-one/9H-Fluoren-9-one (ID 44) exclusively in Orgueil and LEW 85311 suggests a unique oxidative pathway in these meteorites that converts PAHs to their oxygenated derivatives, potentially indicating distinct parent body processing conditions. | IDs 42 and 44 found only in Orgueil and LEW 85311; related to oxidation of PAH structures |
| 4 | The detection of possible terpenes exclusively in soil samples indicates that complex branched isoprenoid structures require enzymatic biosynthesis and are not readily formed through abiotic processes in space, making them reliable biomarkers. | IDs 4, 17, 18 (possible terpenes) found only in soil samples (Iceland Soil, Atacama, Utah soil, GSFC soil) |

Table S8: **Generated hypotheses by AstroAgents powered by Claude Sonnet 3.5** - Iteration 7.

| # | Statement | Key Datapoints |
|---|-----------|----------------|
| 1 | The presence of methylated PAHs (e.g., methylnaphthalenes, trimethylnaphthalenes) predominantly in meteorite samples but not in soil samples suggests that alkylation of PAHs occurs through abiotic processes in space rather than through biological activity. | ID 2 (2-methylnaphthalene), ID 8 (1-methylnaphthalene), ID 15 (trimethylnaphthalene) found mainly in meteorites Orgueil, ALH 83100, Murchison, LEW 85311 |
| 2 | The co-occurrence of organosulfur compounds (dibenzothiophene, trithiolane) specifically in meteorite samples suggests that sulfur incorporation into organic compounds was an important abiotic process in the early solar system, possibly catalyzed by metal sulfides present in the meteorite parent bodies. | ID 14 (dibenzothiophene) in Orgueil, ALH 83100, LEW 85311; ID 27 (1,2,4-trithiolane) in Aguas Zarcas, LEW 85311 |
| 3 | The presence of 1,2,3,4-tetrahydro phenanthrene (ID 36) exclusively in Orgueil and Jbilet Winselwan, along with their unique dimethylpyrene content (ID 45), suggests these meteorites experienced similar parent body processing conditions that were distinct from other carbonaceous chondrites. | ID 36 (1,2,3,4-tetrahydro phenanthrene) and ID 45 (dimethylpyrene) in Orgueil and Jbilet Winselwan; Distribution pattern distinct from other meteorites |
| 4 | The co-occurrence of phenanthrene/anthracene with their oxygenated derivatives (1H-Phenalen-1-one/9H-Fluoren-9-one) specifically in meteorites suggests an abiotic oxidation pathway that operates in space but not in terrestrial environments. | ID 42 (phenanthrene/anthracene) and ID 44 (1H-Phenalen-1-one/9H-Fluoren-9-one) found together in Orgueil and LEW 85311 |

Table S9: **Generated hypotheses by AstroAgents powered by Claude Sonnet 3.5** - Iteration 8.

| # | Statement | Key Datapoints |
|---|-----------|----------------|
| 1 | The presence of methylated PAHs (e.g., methylnaphthalenes, trimethylnaphthalene) predominantly in meteorite samples but not in soil samples suggests that alkylation of PAHs occurs through abiotic processes in space rather than through biological activity. | ID 2 (2-methylnaphthalene), ID 8 (1-methylnaphthalene), ID 15 (trimethylnaphthalene) found mainly in meteorites Orgueil, ALH 83100, Murchison, LEW 85311 |
| 2 | The co-occurrence of organosulfur compounds (dibenzothiophene, trithiolane) specifically in meteorite samples suggests that sulfur incorporation into organic compounds was an important abiotic process in the early solar system, possibly catalyzed by metal sulfides present in the meteorite parent bodies. | ID 14 (dibenzothiophene) in Orgueil, ALH 83100, LEW 85311; ID 27 (1,2,4-trithiolane) in Aguas Zarcas, LEW 85311; notably absent in all soil samples |
| 3 | The presence of high molecular weight PAHs (fluoranthene, pyrene) exclusively in meteorites but not in soil samples indicates that these compounds form through high-temperature gas-phase reactions in space rather than through biological processes. | ID 12 (fluoranthene) and ID 13 (pyrene) found only in meteorites ALH 83100, LON 94101, Orgueil, LEW 85311 |
| 4 | The presence of partially hydrogenated aromatic compounds (indane, acenaphthene, tetrahydrophenanthrene) specifically in meteorites suggests a stepwise PAH formation process in space involving both aromatic ring formation and partial hydrogenation steps. | ID 11 (indane) in ALH 83100, LON 94101, LEW 85311; ID 33 (acenaphthene) in ALH 83100, LEW 85311; ID 36 (tetrahydrophenanthrene) in Orgueil, Jbilet Winselwan |
| 5 | The presence of terpene-like compounds exclusively in soil samples (particularly from extreme environments like Iceland and Atacama) suggests these molecules are reliable biosignatures even in harsh conditions that might be analogous to other planetary environments. | ID 4, ID 17, ID 18 (possible terpenes/sesquiterpenes) found only in soil samples from Iceland, Atacama, Utah, and GSFC |

Table S10: **Generated hypotheses by AstroAgents powered by Claude Sonnet 3.5** - Iteration 9.

| # | Statement | Key Datapoints |
|---|-----------|----------------|
| 1 | The presence of methylated PAHs (e.g., methylnaphthalenes, trimethylnaphthalenes) predominantly in meteorite samples but not in soil samples suggests that alkylation of PAHs occurs through abiotic processes in the early solar system rather than through biological activity. | ID 2,8 (methylnaphthalenes) in Orgueil, ALH 83100, Murchison; ID 15 (trimethylnaphthalene) in Orgueil, Jbilet Winselwan, LEW 85311; Notably absent in soil samples |
| 2 | The co-occurrence of fluoranthene and pyrene (IDs 12,13) exclusively in meteorites, combined with their absence in all soil samples, indicates these 4-ring PAHs are formed through specific high-temperature abiotic processes in space rather than biological processes on Earth. | ID 12 (fluoranthene) in ALH 83100, LON 94101, LEW 85311; ID 13 (pyrene) in Orgueil, LON 94101, LEW 85311 |
| 3 | The presence of organosulfur compounds (1,2,4-trithiolane and dibenzothiophene) exclusively in meteoritic samples suggests that sulfur incorporation into organic compounds occurred through unique chemical pathways in the early solar system, possibly catalyzed by mineral surfaces under reducing conditions. | ID 14 (dibenzothiophene) in Orgueil, ALH 83100, LEW 85311; ID 27 (trithiolane) in Aguas Zarcas, LEW 85311 |
| 4 | The exclusive detection of terpene-like compounds in soil samples but not in meteorites indicates these compounds are reliable biomarkers for distinguishing biological from abiotic organic matter sources. | ID 4,17,18 (possible terpenes) found only in Iceland Soil, Atacama, Utah soil, GSFC soil |
| 5 | The presence of partially hydrogenated PAHs (like indane and acenaphthene) specifically in meteorites suggests a stepwise PAH formation process in space involving hydrogenation steps, rather than direct synthesis of fully aromatic systems. | ID 11 (indane) in ALH 83100, LON 94101, LEW 85311; ID 33 (acenaphthene) in ALH 83100, LEW 85311; ID 36 (tetrahydrophenanthrene) in Orgueil, Jbilet Winselwan |

Table S11: **Generated hypotheses by AstroAgents powered by Claude Sonnet 3.5** - Iteration 10.

| # | Statement | Key Datapoints |
|---|-----------|----------------|
| 1 | The presence of methylated PAHs (e.g., methylnaphthalenes, trimethylnaphthalenes) predominantly in meteorite samples but not in soil samples suggests that abiotic methylation of PAHs occurs through unique chemical pathways in space that are distinct from biological methylation processes on Earth. | ID 2,8 (methylnaphthalenes) found in meteorites Orgueil, ALH 83100, Murchison; ID 15 (trimethylnaphthalene) in Orgueil, Jbilet Winselwan, LEW 85311; Notably absent in most soil samples |
| 2 | The co-occurrence of dibenzothiophene ($C_{12}H_8S$) and 1,2,4-trithiolane ($C_2H_4S_3$) exclusively in meteorite samples suggests a distinct sulfur incorporation pathway in abiotic PAH formation, possibly linked to cosmic ray or UV-induced sulfurization of organic matter in space. | ID 14 (dibenzothiophene) in Orgueil, ALH 83100, LEW 85311; ID 27 (1,2,4-trithiolane) in Aguas Zarcas, LEW 85311 |
| 3 | The presence of high molecular weight PAHs (fluoranthene and pyrene) exclusively in meteorites suggests that abiotic PAH formation in space favors condensation reactions leading to larger ring systems, while biological PAH production on Earth tends toward smaller aromatic compounds. | ID 12 (fluoranthene) and ID 13 (pyrene) found only in meteorites ALH 83100, LON 94101, LEW 85311, Orgueil; Smaller PAHs like ID 7 (toluene) more common in soil samples |
| 4 | The exclusive presence of terpene-like compounds in soil samples but not in meteorites indicates these compounds are reliable biomarkers for distinguishing biological from abiotic organic matter sources. | ID 4, 17, 18 (possible terpenes) found only in biological samples like Iceland Soil, Atacama, Utah soil; absent in all meteorite samples |
| 5 | The co-occurrence of phenanthrene/anthracene with their oxygenated derivatives (1H-Phenalen-1-one/9H-Fluoren-9-one) specifically in meteorites suggests an abiotic oxidation pathway that operates in space but not in terrestrial environments. | ID 42 (phenanthrene/anthracene) and ID 44 (1H-Phenalen-1-one/9H-Fluoren-9-one) found together in Orgueil and LEW 85311 meteorites |

