# OpenReview forum: "AstroAgents: A Multi-Agent AI for Hypothesis Generation from Mass Spectrometry Data"
_ICLR.cc/2025/Workshop/AgenticAI — ICLR 2025 Workshop AgenticAI Oral_

### Official Review · Reviewer_h3RG · 2025-03-03

**Rating:** 6
**Confidence:** 4

**Review:**

Strength:
- The paper presents an agentic framework for automating hypothesis generation from mass spectrometry data.
- Comprehensive experiments evaluate two large language model variants, with thorough human evaluation by domain experts providing credibility to the results.
- This work is clearly presented and easy to follow.

Weakness and comments:
- The paper lacks mathematical formalization of the problem, making it difficult to precisely understand the nature of mass spectrometry data processing and the specific inputs/outputs at each stage of the framework.
- Several critical components are inadequately explained, creating confusion. For instance, Line 149 states "the agent refines its findings based on feedback from the critic agent," but this interaction isn't clearly depicted in Figure 1, leaving the feedback mechanism ambiguous.
- Section 2.4 fails to specify important implementation details, including how many artificial researchers are deployed in the framework and what criteria determine this number, which impacts reproducibility.
- The results section provides insufficient analysis of the presented data. Table 1 contains important findings but is neither properly referenced in the text nor analyzed to explain the implications of the performance metrics, missing an opportunity to derive meaningful insights from the experimental results.

The manuscript can be rated higher if the comments in weakness section are addressed.

---

### Official Review · Reviewer_64zF · 2025-03-04
**AstroAgents: A Multi-Agent AI for Hypothesis Generation from Mass Spectrometry Data**

**Rating:** 8
**Confidence:** 4

**Review:**

This paper present AstroAgents, a multi-agent AI system designed to analyze mass spectrometry data from meteorites and soil samples to generate hypotheses about the emergence of life.

### **Strength:**

1) **Novel application**: The paper demonstrates how LLMs can effectively bridge multiple scientific disciplines (chemistry, astronomy, biology) to generate hypotheses about origins of life.

2) **Clear multi-agent design**: The system's architecture with specialized agents for different tasks is well-conceived, with clear roles and interactions between agents.

3) **Literature-Grounded Reasoning**: The integration of the Literature Review agent that allows hypotheses to be contextually aligned with existing research is promising and interesting.

4) **Concrete examples**: The paper includes several well-documented examples of the hypotheses generated, making the results tangible and understandable.


### **Weakness**:

1) **Novelty assessment**: The determination of whether a hypothesis is "novel" seems subjective, and the paper doesn't clarify how this was established relative to existing literature.

2) **Baseline comparison**: There's no comparison with simpler approaches, making it hard to gauge the real improvement over alternatives

---

### Decision · Program_Chairs · 2025-03-05

Accept (Oral)